# BOOSTING LATENT DIFFUSION MODELS VIA SEMANTIC-DISENTANGLED VAE

## ABSTRACT

Latent Diffusion Models (LDMs) rely on image tokenizers, typically implemented as Variational Autoencoders (VAEs), to compress high-dimensional images into compact latent space, facilitating efficient generative modeling. We contend that VAEs trained solely on pixel-level reconstruction objective struggle to capture rich semantic information, which poses challenges for the modeling of downstream diffusion models. In this paper, we propose that a generation-friendly VAE should have the ability of semantic disentanglement, which means it can encode attribute-level semantic information more effectively. To address this, we introduce **Se**mantic-**d**isentangled VAE (Send-VAE), which leverages the rich semantic knowledge from pre-trained vision foundation models to improve the VAE's ability to disentangle semantics. Specifically, we employ a sophisticated non-linear mapper network to transform VAE's latent representations, then align them with the representations from vision foundation models. The mapper network is designed to bridge the representation gap between VAE and vision foundation models, thus facilitating effective guidance for VAE learning. Additionally, we implement linear probing on attribute prediction tasks to assess the VAE's semantic disentanglement ability, demonstrating a strong correlation with downstream generation performance. Finally, utilizing on the proposed Send-VAE, we train popular flow-based transformers SiTs, and experimental results indicate that our proposed Send-VAE can significantly speed up SiT training and achieves a new state-of-the-art FID score of 1.21 and 1.75 with and without classifier free guidance on ImageNet $256 \times 256$ resolution.

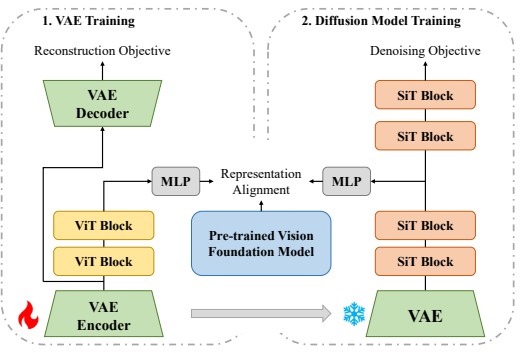
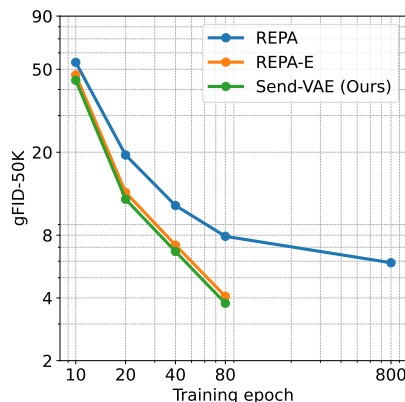

Figure 1: Our Send-VAE aligns the latent representation of VAEs with the semantically rich representation from pre-trained vision foundation models through a specialized mapper network. Unlike the direct alignment employed during diffusion model training, the mapper network can effectively bridge the representation gap, facilitating seamless injection of semantic information. Notably, the usage of Send-VAE results in significantly more efficient and effective training of diffusion models.

# 1 INTRODUCTION

Latent diffusion models (LDMs) Albergo & Vanden-Eijnden (2023); Rombach et al. (2022); Peebles & Xie (2023); Ma et al. (2024) have recently achieved remarkable success in high-resolution image synthesis, establishing new benchmarks in visual fidelity and detail. A critical component of these models is the image tokenizer, which is often implemented using a variational autoencoder (VAE) Kingma & Welling (2013). The VAE compresses input images into structured latent space, thereby reducing the computational demands associated with generating high-resolution images. The performance of the VAE directly influences both the training efficiency and the quality of the output from downstream generation models. Despite its importance, the characteristics of a generation-friendly VAE, which can facilitate effective learning of downstream generation models, remain underexplored.

Traditionally, VAE training emphasizes pixel-level reconstruction, often neglecting alignment with generation objectives. Inspired by REPA Yu et al. (2025), recent studies on VAE Yao et al. (2025); Chen et al. (2025a); Zha et al. (2025) primarily focusing on explicitly aligning the VAE's latent representation with the representation from large-scale, pre-trained visual foundation models such as CLIP Radford et al. (2021) or DINOv2 Oquab et al. (2024). In contrast, REPA-E Leng et al. (2025) extends REPA to an end-to-end joint training strategy through backpropagating the representation alignment loss of diffusion transformers to VAE. Although these approaches have demonstrated significant performance improvements in downstream generation tasks, there is still a lack of explanation regarding what attributes make a VAE generation-friendly.

Inspired by the analysis of 1D tokenizers in Beyer et al. (2025), we hypothesize that the semantic disentanglement ability of VAE is the key factor, which makes the VAE can better encoder attribute-level semantic information. To verify this hypothesis, we first conduct linear probing experiments on attribute prediction benchmarks to measure the semantic disentanglement ability of various VAEs. Strikingly, we observe a strong positive correlation between the linear separability of these attributes within the VAE latent space and the generation quality achieved by the downstream diffusion model. This compelling evidence suggests that the richness and accessibility of attribute-level semantic information is a more fundamental characteristic of a VAE's latent space, conducive to effective diffusion modeling. Consequently, we advocate for the performance on these low-level attribute prediction tasks via linear probing as a novel, more intrinsic metric for evaluating quality of VAE's latent space.

Based on this observation, we propose semantic-disentangled VAE (Send-VAE), which leverages the semantically rich representation from pre-trained vision foundation models to guide the learning of VAE. Unlike previous attempts that directly align the VAE's latent representation with those from vision foundation models, we incorporate a sophisticated non-linear mapper network between VAE and vision foundation models. Such a mapper network targets at bridging the representation gap between VAE and vision foundation models, thus facilitating effective semantic injection to enhance the semantic disentanglement ability of VAE. As shown in Fig. 1 right, when training with flow-based transformers SiTs Ma et al. (2024), Send-VAE can significantly accelerate the SiT training compared with REPA and achieves a new state-of-the-art FID score of 1.21 and 1.75 with and without classifier-free guidance on ImageNet $256 \times 256$ generation.

In summary, this paper makes the following key contributions:

- We propose a VAE with stronger semantic disentanglement ability tends to be a generation-friendly VAE, which can be verified by the strong correlation between linear probing performance on low-level attribute prediction tasks and downstream generation performance.

- To enhance the semantic disentanglement ability of VAE, we propose Send-VAE, a simple yet effective VAE training mechanism through aligning VAE's latent space with vision foundation models using a sophisticated non-linear mapper network.

- Our Send-VAE can significantly accelerate the convergence of diffusion models and achieves a new state-of-the-art FID score on ImageNet 256x256 generation.

## 2 RELATED WORK

**Tokenizers for Image Generation** Image tokenizers are designed to transform high-dimension image inputs into more compact and structured latent representations, facilitating modeling by downstream generative models. These tokenizers can be broadly categorized into continuous and discrete types. Continuous tokenizers, exemplified by Variational Autoencoders (VAEs) Kingma & Welling (2013), are widely adopted in diffusion-based generation models Rombach et al. (2022); Peebles & Xie (2023); Ma et al. (2024); whereas discrete tokenizers, represented by VQGAN Esser et al. (2021), are commonly used in autoregressive (AR) generation models. However, as these tokenizers are typically trained with a pixel-level reconstruction objective, their latent spaces may not be well aligned with the requirements of generation tasks. To address this limitation, recent researches begin to incorporate semantic information into the training of image tokenizers, with the goal of learning latent spaces that are more suitable for generation. For instance, VA-VAE Yao et al. (2025) aligns the latent representations of VAE with pre-trained vision foundation models, significantly improving the generation performance of high-dimensional tokenizers while preserving their original reconstruction capabilities. Inspired by MAE He et al. (2022), MAETok Chen et al. (2025a) incorporates masked image modeling into tokenizer training and leverages multiple target features to learn a semantically rich latent space. Similar strategies have also been explored in discrete tokenizers Xiong et al. (2025); Li et al. (2025). Unlike these explicit alignment-based methods, REPA-E Leng et al. (2025) introduces a end-to-end joint training framework through backpropagating the representation alignment loss of diffusion transformers to VAE. Although REPA-E achieves notable performance gains, its straightforward joint training strategy leaves a fundamental question unanswered: what properties make a VAE well-suited for generation tasks? We propose that a generation-friendly VAE should possess strong semantic disentanglement ability. To this end, we leverage the semantically rich representation from pre-trained vision foundation models to guide the learning process of VAE.

**Diffusion models for image generation.** Diffusion models have emerged as a powerful class of generative models, formulating image synthesis as a progressive denoising process that transforms Gaussian noise into realistic images. Early methods such as DDPM Ho et al. (2020) and DDIM Song et al. (2021) operate directly in the pixel space, requiring numerous iterative steps for high-fidelity generation. To improve efficiency, latent diffusion models (LDMs)Rombach et al. (2022) compress images into a lower-dimensional latent space using pre-trained autoencoders, enabling faster and more scalable training. Most early diffusion modelsNichol & Dhariwal (2021); Rombach et al. (2022) adopt U-Net architectures for noise prediction, while recent advances explore transformer-based designs Peebles & Xie (2023); Ma et al. (2024) to better capture long-range dependencies. In addition to architectural improvements, recent studies have explored leveraging pretrained visual representations to enhance the efficiency and performance of diffusion models, enabling better feature representation and faster convergence. For instance, MaskDiT Zheng et al. (2024) and SD-DiT Zhu et al. (2024) adopt training paradigms from MAE He et al. (2022) and iBOT Zhou et al. to enhance feature learning within the Diffusion Transformer (DiT) framework. REPA Yu et al. (2025) aligns the latent features of a diffusion model with those from a frozen, high-capacity encoder pretrained on large-scale external data, thereby regularizing the generative process. Building upon this idea, SARA Chen et al. (2025b) further introduces structural and adversarial alignment objectives, while SoftREPA Lee et al. (2025) extends the framework to multimodal settings by aligning noisy image representations with soft text embeddings. To avoid reliance on additional pretrained visual models, Dispersive Loss Wang & He (2025) encourages internal representations to disperse in the hidden space, and demonstrates that representation regularization alone can effectively enhance generative modeling. These works explore representation learning of the denoising network within a fixed latent space, while overlooking the representation learning of the VAE.

## 3 METHOD

In this section, we provide a comprehensive introduction to the design of Send-VAE. We begin by analyzing the behavior of three publicly available VAEs including VA-VAE (f16d32) Yao et al. (2025), E2E-VAE Leng et al. (2025), and IN-VAE Leng et al. (2025). We observe that there is a strong correlation between the performance of linear probing on attribute prediction tasks and the downstream generation performance. Based on the analysis, we hypothesize that a generative-

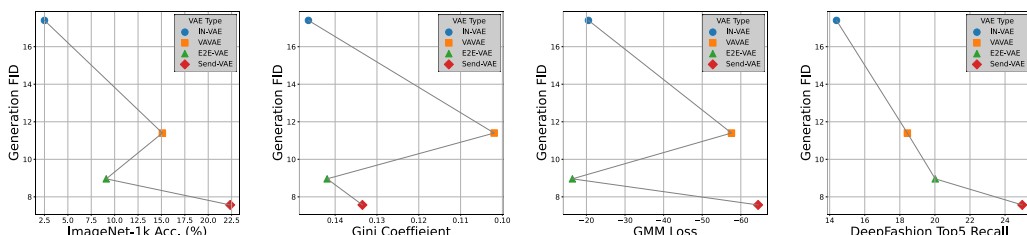

Figure 2: We conduct experiments with three recently proposed evaluation methods for VAE latent space, and show their correlation with down stream generation performance (g-FID). Experimental results on four VAEs with identical specifications indicate that these metrics do not accurately reflect the impact of VAEs on downstream generative performance. Conversely, we find that the ability of VAEs regarding low-level attributes is the key factor.

friendly VAE necessitates a strong semantic disentanglement capability. Thus, we propose Send-VAE, which injects semantic information into VAE through the use of pre-trained vision foundation models. Finally, we regard linear probing on attribute prediction tasks as a measurement of the VAE's semantic disentanglement capability and verify the effectiveness of our Send-VAE.

### 3.1 OBSERVATIONS

To answer the question of what characteristics should a generative-friendly VAE possess, we first investigate the behavior of VAE latent space using three recently proposed evaluation methods, including semantic gap Yu et al. (2025), latent space uniformity Yao et al. (2025), and latent space discrimination Chen et al. (2025a). For semantic gap, the linear probing on ImageNet classification is adopted following REPA Yu et al. (2025). Next, for latent space uniformity, we calculate Gini coefficients of data point distribution using kernel density estimation (KDE) as done in VA-VAE Yao et al. (2025). As for latent space discrimination, we fit a Gaussian mixture model (GMM) into the latent space following MAETok Chen et al. (2025a). We include three publicly available VAEs: VA-VAE (f16d32) Yao et al. (2025), E2E-VAE Leng et al. (2025), IN-VAE Leng et al. (2025) and our Send-VAE, with the final results shown in Fig 2.

**The uniformity and discrimination of latent space are not directly correlated with generation performance.** As shown in Fig 2, we observe that while VA-VAE shows improved uniformity and enhanced downstream generation performance compared with IN-VAE, such a conclusion does not hold true for E2E-VAE. A similar situation also occurs in the evaluation of latent space discrimination. We argue that these metrics only partially reflect the impact of VAEs on generation performance, and cannot accurately describe the characteristics of a generation-friendly VAE.

**The semantic disentanglement ability is the key factor.** Aligning the hidden states of a diffusion model with pretrained vision foundation models is first proposed in REPA Yu et al. (2025) to reduce the semantic gap between them, which has been proven to accelerate the convergence of diffusion models. As for VAEs, we can observe that while directly injecting semantic information can improve generation performance partially (VA-VAE achieves significant performance gains compared with IN-VAE), it is not a necessary requirement for a generation-friendly VAE considering the further performance gains achieved by E2E-VAE. Motivated by the observation in Beyer et al. (2025), we hypothesize that the semantic disentanglement ability of VAE is the key factor and conduct linear probing on attribute prediction tasks to verify it. As show in Fig 2 right, strong correlation between generation performance and the linear probing performance can be observed, which verifies our hypothesis. Meanwhile, our Send-VAE can achieve more powerful semantic disentanglement ability, thus resulting in better generation performance.

### 3.2 SEMANTIC DISENTANGLED VAE

Based on the above hypothesis, we try to enhance the semantic disentanglement ability of VAE and propose our Send-VAE. Specifically, Send-VAE utilizes a sophisticated non-linear mapper network to transform the latent representations of VAE, and aligns the patch-wise transformed representa-

tions with pre-trained vision foundation models. Different from the simple multilayer perceptron (MLP) used in VA-VAE and REPA, our mapper network consists of a patch embedding layer, a stack of vision transformer (ViT) Dosovitskiy et al. (2021) layers, and the final MLP projector. The reason for this is the difference between the training objectives of vision foundation models and VAEs, which leads to a substantial representation gap. Therefore, compared with direct alignment, a sophisticated non-linear mapper network is designed to mitigate the representation gap and enable effective knowledge distillation from semantically rich visual representations to VAE. The overall framework is shown in Fig 1.

Formally, given a clean image $\mathbf{x}$, let $\mathbf{z}$ be the latent representation of $\mathbf{x}$ output by VAE $\mathcal{V}_\theta$, $f$ be a frozen vision foundation model, and $\mathbf{y} = f(\mathbf{x}) \in \mathbb{R}^{N \times D}$ is the encoded representation of $\mathbf{x}$, where $N, D$ are the number of patches and the embedding dimension of $f$, respectively. Following the noise injection mechanism of SiT Ma et al. (2024), PE-VAE first inject random Gaussian noise into $\mathbf{z}$ and get $\mathbf{z}_t$, where $t$ is the time step. Then, the mapper network $h_\phi$ is applied to transform $\mathbf{z}_t$ into $h_\phi(\mathbf{z}_t)$, and the alignment loss can be calculated using patch-wise cosine similarity between $h_\phi(\mathbf{z}_t)$ and $f(\mathbf{x})$:

$$\mathcal{L}_{\text{align}} = \frac{1}{N} \sum_{n=1}^{N} (1 - \frac{h_\phi(\mathbf{z}_t)^{[n]} \cdot f(\mathbf{x})^{[n]}}{\|h_\phi(\mathbf{z}_t)^{[n]}\|\|f(\mathbf{x})^{[n]}\|}), \quad (1)$$

where $n$ is the patch index.

In practice, we use $\mathcal{L}_{\text{align}}$ to finetune a pre-trained VAE for fast convergence. And the original VAE training loss function $\mathcal{L}_{\text{VAE}}$ used in AI (n.d.), is also included, which consists of reconstruction losses ($\mathcal{L}_{\text{MSE}}, \mathcal{L}_{\text{LPIPS}}$), GAN loss ($\mathcal{L}_{GAN}$) and KL divergence loss $\mathcal{L}_{\text{KL}}$. Thus, the overall training objective can be formulated as:

$$\mathcal{L}(\theta, \phi) = \lambda_{\text{align}}\mathcal{L}_{\text{align}} + \mathcal{L}_{\text{VAE}}, \quad (2)$$

where $\theta$ and $\phi$ refer to the parameters of VAE and mapper network.

## 4 EXPERIMENTS

In this section, we conduct comprehensive experiments on the ImageNet dataset Deng et al. (2009) at 256×256 resolution to validate the design choices of Send-VAE, and benchmark its generation performance to demonstrate its superiority over existing approaches.

### 4.1 IMPLEMENTATION DETAILS

We follow the same set up as in REPA-E Leng et al. (2025) unless otherwise specified. All training is conducted on the training split of ImageNet Deng et al. (2009). The data preprocessing protocol is same as in ADM Dhariwal & Nichol (2021) including center-crop and resizing to 256x256 resolution.

**For VAE training**, we train 80 epoch with a global batch size of 1024, AdamW Loshchilov & Hutter (2019) optimizer is adopted and the learning rate is set to $3.0 \times 10^{-4}$. As for the initialization, we experiment with publicly available VAEs, including SD-VAE (f8d4) Rombach et al. (2022), VA-VAE (f16d32) Yao et al. (2025), and IN-VAE (f16d32), which is trained on ImageNet following Rombach et al. (2022). Experimentally, we choose VA-VAE as the default setting. As for alignment loss $\mathcal{L}_{\text{align}}$, we use DINOv2 Oquab et al. (2024) as the vision foundation model, and $\lambda_{\text{align}}$ is set to 1.0.

**For diffusion models**, we choose SiT-XL/1 and SiT-XL/2 for VAEs with 4× and 16× downsampling rates, respectively, where 1 and 2 denote the patch sizes in the transformer embedding layer. We train either 80 epoch or 800 epoch with a global batch size of 256, and gradient clipping and exponential moving average (EMA) are applied stable optimization. The learning rate is set to $1.0 \times 10^{-4}$ and AdamW optimizer is used. REPA loss is also included following the setting in Yu et al. (2025).

**For sampling**, the SDE Euler-Maruyama sampler is used, the number of function evaluations (NFE) is set to 250 by default and the cfg scale is set to 2.5

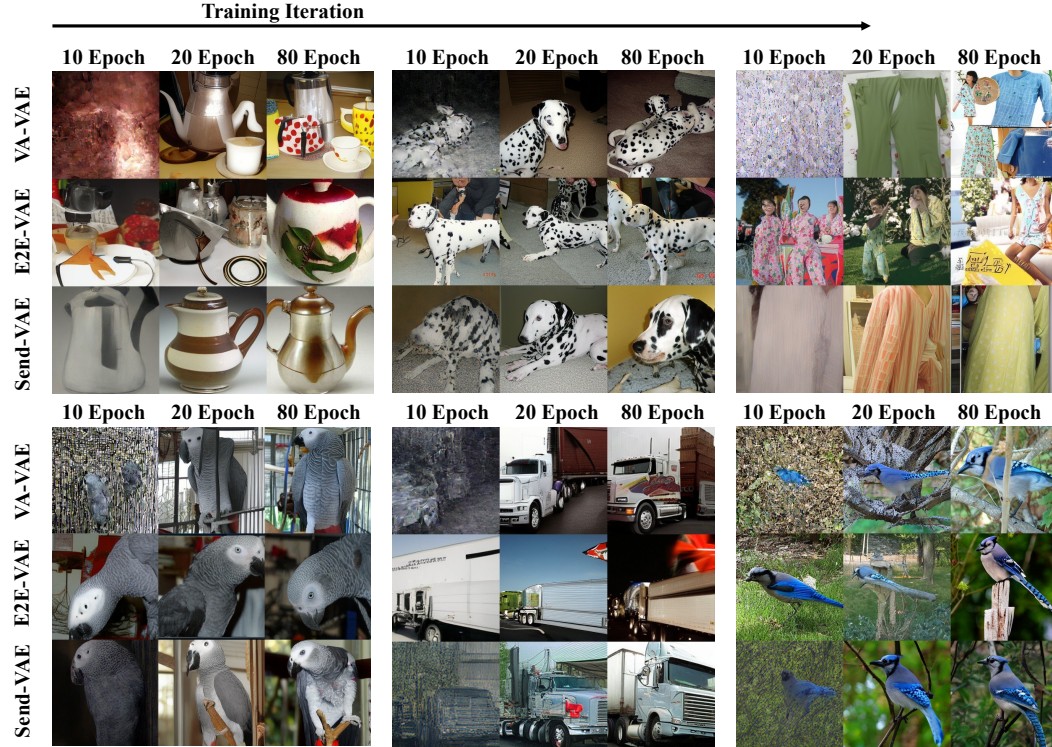

Figure 3: Qualitative comparisons among VA-VAE, E2E-VAE, and Send-VAE. Results for both methods are sampled using the same seed, noise and class label. The classifier-free guidance scale is set to 4.0.

## 4.2 EVALUATION METRICS

For image generation evaluation, we strictly follow the ADM setup Dhariwal & Nichol (2021). Generation quality is assessed using Fréchet Inception Distance (gFID) Heusel et al. (2017), Structural FID (sFID) Nash et al. (2021), Inception Score (IS) Salimans et al. (2016), Precision, and Recall Kynkäänniemi et al. (2019), computed on 50K generated samples. For sampling, we adopt the SDE Euler–Maruyama solver with 250 steps, following the protocols of REPA Yu et al. (2025) and REPA-E Leng et al. (2025). For VAE evaluation, we report reconstruction FID (rFID) on 50K validation images from ImageNet at 256×256 resolution.

## 4.3 SYSTEM-LEVEL COMPARISON ON IMAGENET 256x256 GENERATION

To verify the effectiveness of Send-VAE, we conduct system-level comparison on ImageNet 256x256 conditional and unconditional generation and present the results in Table 1. As we can see, using the same vision foundation model DINOV2, Send-VAE can achieve notable performance gains compared with E2E-VAE and set a new state-of-the-art generation FID score of 1.21 and 1.75 with and without classifier free guidance on ImageNet 256x256 generation. These results highly demonstrate the effectiveness of enhance the semantic disentanglement ability of VAE. Meanwhile, we can notice that Send-VAE can significantly speed up the convergence of diffusion models, evidenced by the superior generation performance (narrowing the gFID score from 3.46 to 2.88 for unconditional generation) when training with only 80 epoch. These results demonstrate that Send-VAE is a generation-friendly VAE, which can facilitate the learning of diffusion models. Meanwhile, some qualitative results are shown in Fig.1 using Send-VAE and SiT-XL/1.

As for reconstruction, we observe that the reconstruction performance of Send-VAE is slightly inferior to that of VA-VAE. We attribute this to the semantic disentangled latent space of Send-VAE, which prevents it from capturing excessive fine-grained low-level details.

Table 1: System-level comparison on ImageNet 256x256 conditional and unconditional generation. Our Send-VAE can significant accelerate the convergence of diffusion models, which achieves a gFID socre of 2.88/1.41 wo/w classifier-free guidance for only 80 epoch of training. Although the performance gap between Send-VAE and E2E-VAE is narrowing when training longer, Send-VAE still achieves further improvements.

| Tokenizer | Method | Training Epoch | #params | rFID | Generation w/o CFG | | | | | Generation w/ CFG | | | | |
|---|---|---|---|---|---|---|---|---|---|---|---|---|---|---|
| | | | | | gFID | sFID | IS | Prec. | Rec. | gFID | sFID | IS | Prec. | Rec. |
| AutoRegressive (AR) | | | | | | | | | | | | | | |
| MaskGiT | MaskGIT Chang et al. (2022) | 555 | 227M | 2.28 | 6.18 | - | 182.1 | 0.80 | 0.51 | - | - | - | - | - |
| VQGAN | LlamaGen Sun et al. (2024) | 300 | 3.1B | 0.59 | 9.38 | 8.24 | 112.9 | 0.69 | 0.67 | 2.18 | 5.97 | 263.3 | 0.81 | 0.58 |
| VQVAE | VAR Tian et al. (2024) | 350 | 2.0B | - | - | - | - | - | - | 1.80 | - | 365.4 | 0.83 | 0.57 |
| LFQ tokenizers | MagViT-v2 Yu et al. (2024) | 1080 | 307M | 1.50 | 3.65 | - | 200.5 | - | - | 1.78 | - | 319.4 | - | - |
| LDM | MAR Li et al. (2024) | 800 | 945M | 0.53 | 2.35 | - | 227.8 | 0.79 | 0.62 | 1.55 | - | 303.7 | 0.81 | 0.62 |
| Latent Diffusion Models (LDM) | | | | | | | | | | | | | | |
| SD-VAE Rombach et al. (2022) | MaskDiT Zheng et al. (2024) | 1600 | 675M | 0.61 | 5.69 | 10.34 | 177.9 | 0.74 | 0.60 | 2.28 | 5.67 | 276.6 | 0.80 | 0.61 |
| | DiT Peebles & Xie (2023) | 1400 | 675M | | 9.62 | 6.85 | 121.5 | 0.67 | 0.67 | 2.27 | 4.60 | 278.2 | 0.83 | 0.57 |
| | SiT Ma et al. (2024) | 1400 | 675M | | 8.61 | 6.32 | 131.7 | 0.68 | 0.67 | 2.06 | 4.50 | 270.3 | 0.82 | 0.59 |
| | FastDiT Yao et al. (2024) | 400 | 675M | | 7.91 | 5.45 | 131.3 | 0.67 | 0.69 | 2.03 | 4.63 | 264.0 | 0.81 | 0.60 |
| | MDT Gao et al. (2023a) | 1300 | 675M | | 6.23 | 5.23 | 143.0 | 0.71 | 0.65 | 1.79 | 4.57 | 283.0 | 0.81 | 0.61 |
| | MDTv2 Gao et al. (2023b) | 1080 | 675M | | - | - | - | - | - | 1.58 | 4.52 | 314.7 | 0.79 | 0.65 |
| | REPA Yu et al. (2025) | 800 | 675M | | 5.90 | 5.73 | 157.8 | 0.70 | 0.69 | 1.42 | 4.70 | 305.7 | 0.80 | 0.65 |
| VA-VAE Yao et al. (2025) | LightingDiT Yao et al. (2025) | 80 | 675M | 0.28 | 4.29 | - | - | - | - | - | - | - | - | - |
| | | 800 | 675M | 0.28 | 2.17 | 4.36 | 205.6 | 0.77 | 0.65 | 1.35 | 4.15 | 295.3 | 0.79 | 0.65 |
| MAETok Chen et al. (2025a) | LightingDiT Yao et al. (2025) | 800 | 675M | 0.48 | 2.21 | - | 208.3 | - | - | 1.73 | - | 308.4 | - | - |
| E2E-VAE Leng et al. (2025) | REPA Yu et al. (2025) | 80 | 675M | 0.28 | 3.46 | 4.17 | 159.8 | 0.77 | 0.63 | 1.67 | 4.12 | 266.3 | 0.80 | 0.63 |
| | | 800 | 675M | | 1.83 | **4.22** | 217.3 | 0.77 | 0.66 | 1.26 | 4.11 | 314.9 | 0.79 | 0.66 |
| Send-VAE | REPA Yu et al. (2025) | 80 | 675M | 0.31 | 2.88 | 4.67 | 175.3 | 0.78 | 0.62 | 1.41 | 4.41 | 301.7 | 0.79 | 0.65 |
| | | 800 | 675M | | **1.75** | 4.41 | 218.57 | **0.79** | 0.64 | **1.21** | **4.10** | **315.1** | **0.79** | **0.66** |

Table 2: Ablation on the depth of mapper network.

| Depth | gFID↓ | sFID↓ | IS↑ | Prec.↑ | Rec.↑ |
|---|---|---|---|---|---|
| 0 | 9.20 | 7.06 | 104.2 | 0.73 | 0.57 |
| 1 | **8.42** | **5.05** | **108.3** | **0.74** | **0.60** |
| 2 | 9.47 | 5.33 | 100.4 | 0.73 | **0.60** |

Table 3: Ablation on noise injection.

| Noise Injection | gFID↓ | sFID↓ | IS↑ | Prec.↑ | Rec.↑ |
|---|---|---|---|---|---|
| ✗ | 8.42 | 5.05 | 108.3 | 0.74 | 0.60 |
| ✓ | **7.57** | 5.37 | 115.3 | **0.74** | **0.60** |

Besides, we also provide qualitative comparisons among VA-VAE, E2E-VAE and Send-VAE in Fig 3 We generates images from the same label and initial noise using checkpoints trained by 10 epoch, 20 epoch, and 80 epoch, respectively. As we can see, training diffusion models using Send-VAE demonstrate superior image generation quality compared to VA-VAE and E2E-VAE. Meanwhile, Send-VAE can significantly speed up the training process of diffusion models, evidenced by the more structurally meaningful images during early stages of training process. Some visualization results are presented in Fig 4 to show that training diffusion models with Send-VAE can generate high-quality images.

## 4.4 ABLATION STUDIES

In this section, we provide detailed ablation studies to demonstrate the effectiveness of each design in Send-VAE. Unless otherwise specified, we train a SiT-B/1 with REPA loss for 80 epoch, and report the downstream unconditional generation performance.

**Ablation on Depth of Mapper Network.** We ablate the depth of our proposed mapper network to analyze its impact on downstream generation performance. As shown in Table 3, a mapper with one

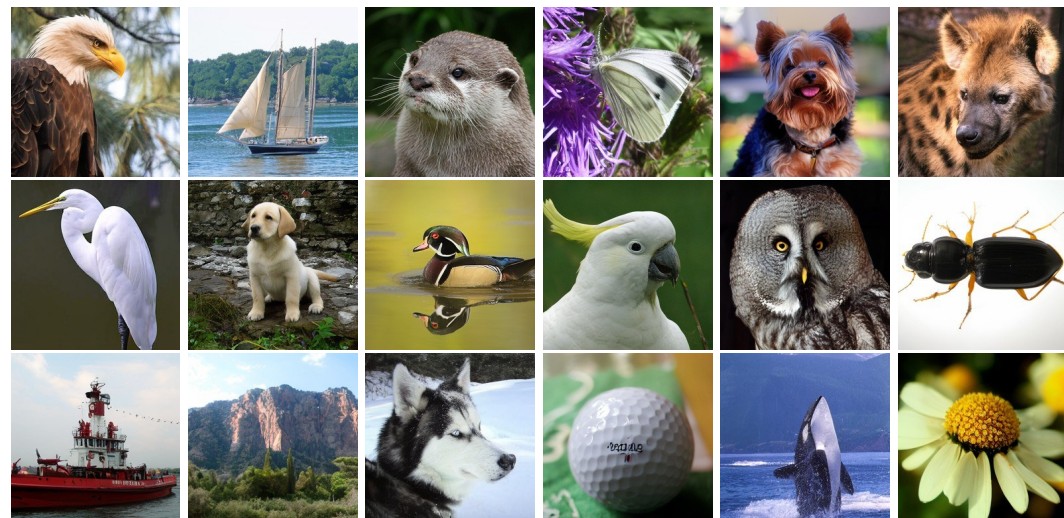

Figure 4: Qualitative Results on ImageNet 256 × 256 using Send-VAE and SiT-XL.

Table 4: Ablation on different vision foundation models (VFMs)

| VFMs | gFID↓ | sFID↓ | IS↑ | Prec.↑ | Rec.↑ |
|------|-------|-------|-----|--------|-------|
| CLIP | 9.85 | 5.59 | 100.8 | 0.71 | 0.62 |
| I-JEPA | 9.70 | 5.40 | 102.9 | 0.72 | 0.60 |
| DINOv2 | 7.57 | 5.37 | 115.3 | 0.74 | 0.60 |
| DINOv3 | 7.16 | 5.57 | 125.3 | 0.75 | 0.58 |

layer of ViT achieves the best performance (gFID=8.42), outperforming both shallower (0 layer) and deeper (2 layer) configurations. We argue that the insufficient capacity of shallow mapper fails to bridge the representation gap between VAE and visual foundation models, resulting in a decrease in the semantic disentanglement ability of VAE. While for the deeper one, it weaken the foundational model's impacts on VAE due to the stronger fitting capability. Such experimental results demonstrate the necessity of employing a mapper network to bridge representation gap, which can facilitate effective semantic injection.

**Ablation on Injecting Noise to Latent Representations.** Table 3 presents the ablation results of injecting noise to latent representations. As we can see, injecting noise during the alignment process can bring significant performance gains. We attribute its effectiveness to a form of data augmentation, which ensures that even with noise injected, the latent representation extracted by the VAE retains rich disentangled semantic information, making it better suited for the denoising process of the downstream diffusion model.

**Ablation on Vision Foundation Models.** We also investigate the influence of vision foundation models and present the ablation results in Table 2. Specifically, we include four types of vision foundation models, including CLIP Radford et al. (2021), I-JEPA Assran et al. (2023), DINOv2 Oquab et al. (2024), and DINOv3 Siméoni et al. (2025). As we can see, regardless of the type of vision foundation models, adding $\mathcal{L}_{\text{align}}$ consistently improve the generation performance of diffusion models. Among them, the DINO family (DINOv2 and DINOv3) achieves the best performance, which is consistent with the findings of REPA and REPA-E. We argue that the object-centric features of DINO can more effectively facilitate the VAE in learning a semantic disentangled latent space, thus resulting in superior generation performance.

**Ablation on the Initialization of VAE.** To demonstrate the generalization of our method to various VAE initialization, we conducted experiments on three commonly used VAEs, including SD-

Table 5: Ablation on the Initialization of VAE.

| VAE Initialization | gFID↓ | sFID↓ | IS↑ | Prec.↑ | Rec.↑ |
|---|---|---|---|---|---|
| SD-VAE | 21.41 | 5.30 | 65.0 | 0.62 | **0.63** |
| $+\mathcal{L}_{\text{align}}$ | **11.86** | **5.25** | **95.2** | **0.73** | 0.58 |
| IN-VAE | 17.43 | 5.93 | 72.7 | 0.64 | 0.63 |
| $+\mathcal{L}_{\text{align}}$ | **8.25** | **4.68** | **105.2** | **0.74** | **0.60** |
| VA-VAE | 11.40 | 6.58 | 93.5 | 0.71 | 0.59 |
| $+\mathcal{L}_{\text{align}}$ | **7.57** | **5.37** | **115.3** | **0.74** | **0.60** |

VAE AI (n.d.), IN-VAE Leng et al. (2025) and VA-VAE Yao et al. (2025). The results are shown in Table. 1. As we can see, across all variations, our $\mathcal{L}_{\text{align}}$ can consistently improves final generation performance, which demonstrates that insensitiveness of our method to the VAE initialization.

## 4.5 MEASUREMENT OF SEMANTIC DISENTANGLEMENT ABILITY

To give a system-level measurement of semantic disentanglement capability, we adopt linear probing on attribute prediction benchmarks across distinct domains to measure the semantic disentanglement ability of various VAEs. Specifically, three attribute prediction benchmarks are used to ensure a comprehensive evaluation, including CelebA Liu et al. (2015), DeepFashion Liu et al. (2016) and AwA Lampert et al. (2013). We conduct linear probing on the flattened latent representation from VAE encoder and show the results in Table 6. As we can see, among all benchmarks, the performance of attribute prediction is positively correlated with the down-stream generation performance. These results strongly support our hypothesis, and making the linear probing on attribute prediction task a suitable metric to evaluate the goodness of a VAE for diffusion. Meanwhile, we observe that Send-VAE can significantly enhance the semantic disentanglement ability of VAE and achieve superior generation performance.

Table 6: System-level measurement of semantic disentanglement ability of various VAEs. F1 score is adopted for all benchmarks.

| Benchmarks | IN-VAE | VA-VAE | E2E-VAE | Send-VAE |
|---|---|---|---|---|
| CelebA | 0.6222 | 0.6347 | 0.6439 | 0.6647 |
| DeepFasion | 0.0786 | 0.1094 | 0.1177 | 0.1385 |
| AwA | 0.5567 | 0.5948 | 0.6441 | 0.6623 |
| gFID | 17.43 | 11.40 | 8.96 | 7.57 |

## 5 CONCLUSION

In this paper, we try to answer the question: what properties make a VAE generation-friendly. We hypothesize that the semantic disentanglement ability of VAE is the key factor, which is verified by the strong correlation between the linear separability of low-level attributes within VAE latent space and the generation performance. This prompts us to utilize the rich semantic representation from pre-trained vision foundation models to enhance the semantic disentanglement ability of VAE. In detail, we propose Send-VAE through aligning the VAE latent representations with those from pre-trained vision foundation models with the use of a sophisticated non-linear mapper network. Such a mapper can bridge the representation gap between VAE and foundation models, thus facilitating effective semantic injection. Experimental results on ImageNet 256x256 generation indicate that our Send-VAE can significantly speed up the training of diffusion models and achieve a new state-of-the-art generation performance.

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

# A APPENDIX

## A.1 LLM USAGE

LLMs are only used to meticulously refine the draft by correcting grammatical errors and improving sentence fluency. During the conceptualization and research design phases of the paper, we do not rely on LLMs; all research ideas and innovations are independently developed by our team.

