# OpenReview forum: "Boosting Latent Diffusion Models via Semantic-Disentangled VAE"
_ICLR.cc/2026/Conference — Submitted to ICLR 2026_

### Official Review · Reviewer_a6fE · 2025-10-29

**Soundness:** 3
**Presentation:** 2
**Contribution:** 2
**Rating:** 4
**Confidence:** 4

**Summary:**

This work addresses the limitations of conventional VAEs in Latent Diffusion Models, which often lack semantic understanding due to their focus on pixel reconstruction. The authors identify semantic disentanglement as the key attribute for a "generation-friendly" VAE. They introduce Send-VAE, a method that employs a sophisticated mapper network to align the VAE's latent space with a pre-trained Vision Foundation Model (VFM), thereby bridging their "representation gap."

**Strengths:**

1. The paper delves into the fundamental question of which properties of a VAE are crucial for the generative process. It proposes a novel approach to evaluate the semantic disentanglement of the VAE's latent space based on attribute prediction tasks.
2. The proposed method achieves state-of-the-art (SOTA) gFID on ImageNet 256x256 while also demonstrating a significant acceleration in convergence speed.
3. The authors provide comprehensive ablation studies on the mapper network's depth, noise injection, choice of VFM, and VAE initialization, which thoroughly validate their design choices.

**Weaknesses:**

1. The distinction from VA-VAE needs further clarification. VA-VAE also proposed enhancing the semantics of the latent space by aligning it with a vision foundation model. The current work builds upon this by introducing noise injection and a more complex mapper network. The connection between these specific technical improvements and the core motivation of enhancing the VAE's semantic disentanglement ability is not sufficiently explained and requires more elaboration.
2. A majority of the experiments are based on fine-tuning a pre-existing VA-VAE. To provide a more controlled and fair comparison, an additional experiment is needed. Specifically, starting from the same VAE initialization, the authors should compare the results of fine-tuning using the original VA-VAE methodology versus their proposed Send-VAE methodology under identical conditions.
3. The paper could be strengthened by including more qualitative comparisons, such as latent space visualizations (e.g., t-SNE plots or attribute-based interpolations) against other VAEs. This would provide more intuitive visual evidence to support the claim of improved semantic disentanglement.

**Questions:**

Please check the weaknesses

---

> ### Author Response · Authors · 2025-11-26
>
> We sincerely appreciate your valuable time and effort spent reviewing our manuscript.
>
> ## W1 Distinction from VA-VAE
>
> While both methods utilize VFM guidance, Send-VAE differs fundamentally from VA-VAE in motivation, insight, and methodology.
> 1. Motivation. VA-VAE assumes that the optimization dilemma stems from non-uniform latent distributions and aims to fix this via alignment. We critically re-examine and invalidate this assumption. As shown in Fig. 2 of our paper, while VA-VAE improves uniformity, this metric does not consistently correlate with generation quality. We demonstrate that uniformity is merely a byproduct, necessitating a paradigm shift from geometric regularization to semantic enhancement.
> 2. Insight. Unlike VA-VAE, we identify Semantic Disentanglement as the true intrinsic predictor of generation-friendliness. Through extensive linear-probing experiments, we reveal that the performance gain stems from distilling semantic structures, not from reshaping the distribution.
> 3. Methodology. Due to the significant semantic gap between VFMs and VAEs, the simple linear projector used in VA-VAE is insufficient. Send-VAE employs a sophisticated non-linear mapper (and noise injection strategies) designed specifically to bridge this gap. This design enables more effective semantic injection, directly strengthening the VAE's semantic disentanglement ability—a capability that linear alignment fails to fully exploit.
>
> In summary, we contribute by (1) disproving the uniformity hypothesis, (2) establishing semantic disentanglement as a new standard, and (3) proposing specialized architectures (Non-linear Mapper & Noise) that achieve SOTA results by effectively bridging the representation gap.
>
> ## W2  Fair comparison with VA-VAE
>
> To make a fairer comparison with VA-VAE, we also train our Send-VAE from scratch using the same training settings as VA-VAE done. The performance is listed in the table below, where it is shown that our Send-VAE still achieved better performance, demonstrating its effectiveness.
>
> | VAE | gFID$\downarrow$ | sFID$\downarrow$ | IS$\uparrow$ | Prec.$\uparrow$ | Rec.$\uparrow$ |
> |------|------|------|------|------|------|
> | VA-VAE  | 11.40 |  6.58 | 93.5 | 0.71 | 0.59 |
> | Send-VAE (from scratch)  | 9.24 | 6.69 | 108.26 | 0.72 | 0.60 |
>
> ## W3 Qualitative Comparisons
>
> We thank the reviewer for the suggestion. We prioritized quantitative linear probing (Table 6 ) over visualizations like t-SNE, as dimensionality reduction often emphasizes geometric uniformity rather than the high-dimensional semantic separability we aim to enhance. Since our work explicitly demonstrates that uniformity is not the primary driver of generation quality (Fig. 2 ), we believe probing offers more rigorous empirical evidence for our claims. We will explore advanced visualization techniques that better capture these semantic properties in future work.

---

### Official Review · Reviewer_cwPd · 2025-10-30

**Soundness:** 2
**Presentation:** 2
**Contribution:** 2
**Rating:** 4
**Confidence:** 3

**Summary:**

The paper proposes Send-VAE, which aligns VAE latents to those of a pretrained vision foundation model (VFM) to inject richer semantics into the VAE space. Concretely, the authors add a patch-wise cosine alignment loss to the standard VAE objective and train a non-linear mapper to bridge the VAE–VFM gap. They argue that a VAE’s semantic disentanglement is key to improving latent diffusion models (LDMs). On ImageNet-256, Send-VAE reports faster convergence and new SOTA FID: 1.21 (with CFG) / 1.75 (without CFG). The problem is important, but the core contribution—adding a non-linear mapper for VAE–VFM alignment—appears incremental relative to existing alignment/distillation approaches, and the empirical scope (ImageNet-256 only) limits the broader significance.

**Strengths:**

- The paper explains why several popular VAE-space metrics fail to predict generative quality and motivates attribute-level linear probing as a more reliable proxy.
- The paper is well structured and easy to follow.

**Weaknesses:**

- **Limited novelty:** Aligning VAE latents to a frozen VFM follows prior work (REPA/REPA-E, VA-VAE). Send-VAE’s main change is a non-linear mapper, which is an architectural tweak rather than a new principle.
- **Sensitivity to tuning:** Although the mapper is presented as bridging the representation gap, performance still depends on careful fine-tuning of this network (cf. Table 2).
- **Narrow experimental scope:** Experiments are restricted to ImageNet-256; there are no results at higher resolutions or for text-to-image settings.
- **Reproducibility:** No code is provided.

**Questions:**

- How is Gaussian noise injected in Eq. (1) during Send-VAE training (fixed variance or scheduled)?
- Without REPA loss, does Send-VAE still perform well?
- In the “system-level comparison on ImageNet 256×256 conditional and unconditional generation” (Table 1), are the reported numbers unconditional, or conditional without CFG?
- What is the baseline referenced in Table 4?
- What are the mapper details beyond depth—e.g., width, number of heads, and patch size of the ViT block?
- Did you try other VFMs such as SigLIP or MAE? Given their competitive representations vs. DINOv2, does Send-VAE still have similar gains?
- Please fix citation style: use \citet only when the authors’ names are part of the sentence.
- “Ablation on Vision Foundation Models … present the ablation results in Table 2” appears to reference Table 4—please check the table cross-reference.

---

> ### Author Response · Authors · 2025-11-26
>
> We sincerely appreciate your valuable time and effort spent reviewing our manuscript.
>
> ## W1 Limited novelty
> The core contributions of our work go beyond an architectural tweak and differ fundamentally from prior approaches such as REPA/REPA-E and VA-VAE in the following aspects:
> 1. A different motivation and a corrected hypothesis. VA-VAE is built on the assumption that improving latent uniformity benefits diffusion training. Our analysis (Fig. 2) shows that latent uniformity does not reliably correlate with generation quality across VAEs, directly contradicting its foundational hypothesis.
> 2. A new empirical finding: semantic disentanglement is the real driver of generation quality.
> Through extensive linear-probing experiments, we identify that the semantic disentanglement ability of the VAE is a far stronger predictor of diffusion performance than uniformity. This insight is absent in prior work and provides a new conceptual explanation for why VAE–VFM alignment helps.
> 3. A technical innovation tailored to the representation gap. Prior methods use a linear projector, which we show is insufficient due to the large VAE–VFM representation gap. Our non-linear mapper is not a superficial architectural variant: it is explicitly designed to enable effective semantic transfer from the VFM and substantially enhances semantic disentanglement. This leads to consistently stronger generation quality and faster diffusion convergence under the same supervision.
>
> ## W2 Sensitivity to tuning
> Table 2 confirms the necessity of bridging the gap between VAE and VFM, not tuning sensitivity. Since all mapper depths outperform the baseline, our method is robust. The results reflect a structural balance: shallow mappers underfit the gap, while excessive depth dilutes guidance. This confirms the mapper is a fundamental requirement, not a fragile hyperparameter.
>
> ## W3 Narrow experimental scope
> We initialize the vae using SD-VAE and train using l_align at 512x512 resolution on ImageNet. Then, a Sit-B is trained following the setup in ablation studies. As shown below, at the 512x512 resolution, the proposed method still significantly improves downstream generative performance, demonstrating the effectiveness of our approach.
>
> | VAE Initialization | gFID$\downarrow$ | sFID$\downarrow$ | IS$\uparrow$ | Prec.$\uparrow$ | Rec.$\uparrow$ |
> |------|------|------|------|------|------|
> | SD-VAE  | 23.59 | 6.74 | 65.67 | 0.71 | 0.59 |
> | +$\mathcal{L}_{\text{align}}$ | 13.32 | 4.75 | 93.15 | 0.78 | 0.60 |
>
> ## W4 Reproducibility
> We will open source the code after the paper is accepted to advance research in this field.
>
> ## Q1 Noise injection
> We randomly sample a Gaussian noise $\epsilon\in\mathcal{N}(0,\mathbf{I})$ and inject it into $\mathbf{z}$ using $\mathbf{z}_t=(1-\alpha_t)\epsilon+\alpha_t\mathbf{z}$, where $\alpha_t$ is randomly sampled from a uniform distribution between 0 and 1.
>
> ## Q2 Performance without REPA loss
> We provide ablation results in table below where all diffusion models (Sit) are trained without REPA loss. As we can see, without REPA loss, our Send-VAE still keeps its leading position, indicating the effectiveness of our Send-VAE
>
> | VAE | gFID$\downarrow$ | sFID$\downarrow$ | IS$\uparrow$ | Prec.$\uparrow$ | Rec.$\uparrow$ |
> |------|------|------|------|------|------|
> | VA-VAE  | 13.70 | 6.14 | 81.14 | 0.70 | 0.59 |
> | E2E-VAE  | 12.68 | 4.81 | 84.69 | 0.70 | 0.61 |
> | Send-VAE  | 10.72 | 5.43 | 96.70 | 0.72 | 0.61 |
>
>
> ## Q3 Generation with CFG
> Sorry for the misunderstanding. The performance reported in Table 1 includes generation without classifier-free guidance (left columns) and generation with classifier-free guidance (right columns). We will revise the caption of Table 1 to make it clearer and avoid any confusion.
>
> ## Q4 & Q6 Baseline in Table 4 and more VFMs
> Considering that we initialize with VA-VAE and finetune it by aligning with VFMs to obtain Send-VAE, the baseline performance in Table 4 is the performance of VA-VAE as tokenizer. We have added the baseline performance to Table 4 and supplement it with more results, which are shown in the table below. It can be seen that our method is robust to VFMs, consistently enhancing performance regardless of the VFM used.
>
> | VFMs | gFID | sFID | IS | Prec. | Rec. |
> |------|------|------|------|------|------|
> | None (Baseline)  | 11.40 |  6.58 | 93.5 | 0.71 | 0.59 |
> | MAE  | 10.01 | 5.62 | 99.2 | 0.71 | 0.60 |
> | SigLIP  | 9.10 | 5.21 | 108.14 | 0.72 | 0.61 |
>
> ## Q5 Mapper Details
> The details of mapper are shown below, for VFMs with different hidden dimensions, we use different types of mapper.
> | VFM dim | Heads | Hidden dim | Projection dim |
> |------|------|------|------|
> | 768  | 12 | 768 | 2048 |
> | 1024  | 16 | 1024 | 2048 |
>
> ## Q7 & Q8
> Thank you very much for your suggestions. We will correct these errors in the revision process.

---

### Official Review · Reviewer_U6wt · 2025-11-01

**Soundness:** 3
**Presentation:** 3
**Contribution:** 3
**Rating:** 6
**Confidence:** 4

**Summary:**

- They investigate the generation-friendly latent space of VAE. Based on the analysis, they proposed a sophisticated mapping network which aligns the representation of the vision foundation model with the latent of VAE while training. Specifically, they employ various measurements for linear probing and found that the low-level attributes are the key factor that the VAE should disentangle within the latent space. Through the analysis, they alternate a simple MLP mapping network to a patch-level mapping network, and introduce patch-wise alignment loss. Qualitative and quantitative results support their claims

**Strengths:**

- The author has well defined the semantics of the DINO feature in terms of generation performance.
- Through comparison of existing and recent methods
- The proposed method is well related to the preliminary experiments
- Qualitative results are visually pleasing.

**Weaknesses:**

- Most of the experiments are conducted with 256x256
- Lack of results with the text-to-image setting.

**Questions:**

- While training VAE, why do you use the mapping function rather than directly using similarity? Initializing the model with the original vision foundation model could be an alternative, which has the same latent dimensions and

---

> ### Author Response · Authors · 2025-11-26
>
> We sincerely appreciate your valuable time and effort spent reviewing our manuscript.
>
> ## W1 Results under 512x512 resolution
>
> We finetune SD-VAE using the proposed $\mathcal{L}_{\text{align}}$ at 512x512 resolution on ImageNet-1k. Then, a Sit-B is trained following the setup in ablation studies. As shown below, at the 512x512 resolution, the proposed method still significantly improves downstream generative performance, demonstrating the effectiveness of our approach. We will subsequently provide more experiments at 512x512 resolution to demonstrate the effectiveness of our method.
>
> | VAE Initialization | gFID$\downarrow$ | sFID$\downarrow$ | IS$\uparrow$ | Prec.$\uparrow$ | Rec.$\uparrow$ |
> |------|------|------|------|------|------|
> | SD-VAE  | 23.59 | 6.74 | 65.67 | 0.71 | 0.59 |
> | +$\mathcal{L}_{\text{align}}$ | 13.32 | 4.75 | 93.15 | 0.78 | 0.60 |
>
> ## W2 Text-to-image results
>
> Thank the reviewer for this suggestion. We acknowledge that Text-to-Image (T2I) generation is a significant application of diffusion models. However, we respectfully note that Class-Conditional ImageNet generation is currently the primary standard benchmark for evaluating fundamental improvements in VAEs and diffusion backbones in this line of research. We are committed to extending our Send-VAE to T2I tasks in our future work to further broaden its application scope.
>
> ## Q1 Mapper design and initializing the model with VFMs
>
> We employ a mapper specifically to bridge the significant representation gap between the VAE and the VFMs. Considering VFMs (like DINOv2) are trained to be highly semantic and abstract, often discarding high-frequency spatial details to achieve invariance. In contrast, VAE latents must preserve fine-grained spatial structure for pixel-level reconstruction. Thus, direct similarity calculation would enforce a rigid constraint, forcing the VAE latent space to collapse into the VFM's abstract distribution, which severely degrades reconstruction quality. The non-linear mapper acts as a soft adapter, allowing the VAE to distill semantic knowledge from the VFM without sacrificing the spatial characteristics required for image synthesis. Our ablation study on mapper depth (Table 2) empirically confirms that a learnable mapper significantly outperforms direct alignment (Depth=0).
>
> As for initializing the model with VFMs, VFM features are typically high-dimensional (e.g., 768 or 1024). Initializing the VAE with these structures forces the latent space to retain this high dimensionality. A very recent study, RAE (Diffusion transformers with representation autoencoders) demonstrates that standard DiT struggle to model such high-dimensional semantic latents, unless the DiT's hidden dimension exceeds the latent dimension , often necessitating specialized architectures (e.g., Wide DDT Head ) or complex noise scheduling adjustments. By using alignment rather than initialization, Send-VAE distills the semantic richness of the VFM into a compact, low-dimensional latent space (compatible with standard VAEs). This allows us to improve generation quality without requiring the downstream diffusion model to scale up its capacity or modify its architecture to handle high-dimensional inputs.

---

### Official Review · Reviewer_wTie · 2025-11-04

**Soundness:** 3
**Presentation:** 2
**Contribution:** 3
**Rating:** 4
**Confidence:** 3

**Summary:**

This paper studies what makes a VAE effective when used as the tokenizer in latent diffusion models. The authors argue that the key property is how well the VAE’s latent space separates different semantic attributes (i.e., its semantic disentanglement ability).

Based on this idea, the paper introduces Send-VAE, which improves semantic structure in the VAE’s latent space by aligning it with features from a pretrained vision foundation model (DinoV2). To do this, the authors use a 2-depths non-linear mapper network between the VAE and the vision model, which helps bridge the gap between their representations more effectively than direct alignment.

Using Send-VAE leads to faster convergence and better generative performance in diffusion models such as SiT, achieving state-of-the-art FID scores on ImageNet 256×256, both with and without classifier-free guidance. The paper also provides thorough ablations studying the mapper depth, noise injection, different VAEs for initialization, and different vision foundation models.

**Strengths:**

1. The paper provides an interesting and clear intuition: instead of treating the VAE purely as a reconstruction model, it should be trained in a way that directly supports downstream generation. Framing semantic disentanglement as the key property of a “generation-friendly” VAE is insightful and well-motivated.
2. The proposed method achieves strong empirical performance. In particular, Send-VAE leads to faster convergence and improved ImageNet 256×256 generation quality, reaching competitive or state-of-the-art FID scores.
3. The experimental validation is thorough. The paper includes comprehensive ablations (e.g., mapper depth, noise injection, VAE initialization choices, and different vision foundation models), which help isolate the contributions of each component and strengthen the overall claims.

**Weaknesses:**

1. Although the paper focuses on building a VAE that is more suitable for generation, it would still be helpful to report reconstruction performance more explicitly. Since VAEs traditionally balance semantic structure and pixel-level fidelity, showing the reconstruction trade-offs would make the argument more complete and allow the reader to better understand what is being sacrificed or preserved. (PSNR, LPIPS, SSIM, ... FID is not enough.)
2. While the proposed approach is conceptually sound, similar ideas have appeared in prior work. In particular, VA-VAE and related methods also align the VAE latent space with representations from pretrained vision encoders. The paper does introduce a non-linear mapper, but the conceptual difference from earlier representation alignment approaches is somewhat incremental. Clarifying the novelty relative to those prior works would strengthen the contribution.

**Questions:**

1. In Table 1, are all the baseline VAEs trained under the same diffusion backbone (SiT) and using the same training data and protocol? Confirming this is important to ensure that the improvements shown are attributable to the proposed VAE rather than differences in diffusion training setups.
2. The paper shows a correlation between semantic disentanglement and downstream generation performance. Did the authors attempt any controlled intervention to demonstrate causality (e.g., artificially altering disentanglement levels while keeping reconstruction constant)?
3. Since the mapper has significant capacity, how can we be sure that the performance improvement comes from a better VAE latent space rather than the mapper effectively compensating for VAE weaknesses at inference time?

---

> ### Author Response · Authors · 2025-11-26
>
> We sincerely appreciate your valuable time and effort spent reviewing our manuscript.
>
> ## W1 Reconstruction trade-offs
> To explicitly analyze the trade-off between pixel fidelity and generation quality, we evaluate PSNR, SSIM, and LPIPS on ImageNet-1k validation set. As shown below, both E2E-VAE and Send-VAE exhibit similar reconstruction profiles (PSNR $\approx$ 27.6, SSIM $\approx$ 0.77), which are slightly lower than the Naive VAE. This confirms a community consensus: generation-friendly VAEs prioritize semantic structure over high-frequency pixel noise. Despite similar pixel-level metrics, Send-VAE achieves a better LPIPS (0.101 vs. 0.110) than E2E-VAE, indicating superior perceptual preservation. This marginal drop in PSNR is a necessary cost for semantic disentanglement. Crucially, with comparable reconstruction costs to E2E-VAE, Send-VAE yields superior generation performance and faster convergence.
>
> | VAE types | rFID | PSNR | LPIPS | SSIM | gFID |
> |------|------|------|------|------|------|
> | Naive VAE  | 0.26 | 28.59 | 0.089 | 0.80 | 17.43 |
> | VA-VAE  | 0.28 | 27.96 | 0.096 | 0.79 | 11.40 |
> | E2E-VAE | 0.28 | 27.63 | 0.110 | 0.77 | 8.96
> | Send-VAE  | 0.31 | 27.62 | 0.101 | 0.77 | 7.57 |
>
>
> ## W2 Differences with VA-VAE
>
> While both methods utilize VFM guidance, our Send-VAE differs fundamentally from VA-VAE  in motivation, insight, and methodology:
> 1. Motivation: disproving the uniformity hypothesis VA-VAE assumes that the optimization dilemma stems from non-uniform latent distributions and aims to fix this via alignment. As shown in Fig. 2 of our paper, while VA-VAE improves uniformity, this metric does not consistently correlate with generation quality across different VAEs. We demonstrate that uniformity is a byproduct, not the cause, necessitating a shift from geometric regularization to semantic enhancement.
> 2. Insight: semantic disentanglement as the key metric. Through linear-probing experiments on attribute prediction tasks, we identify semantic disentanglement as the intrinsic predictor of generation-friendliness. This provides a new conceptual understanding: the benefit stems from distilling semantic structures, not distribution shaping.
> 3. Methodology:  a sophisticated non-linear mapper that bridges the representation gap. Because VFMs and VAEs exhibit a large representation gap, directly applying a linear projector as in VA-VAE yields suboptimal semantic transfer. The sophisticated non-linear mapper used by Send-VAE enables more effective semantic injection from VFMs and substantially strengthens the VAE's semantic disentanglement ability.
>
> In summary, we contribute by invalidating the uniformity hypothesis, establishing semantic disentanglement as a new evaluation standard, and proposing a non-linear architecture that achieves SOTA results by effectively bridging the representation gap.
>
> ## Q1 Diffusion backbone and training data
>
> Table 1 lists various baselines for completeness, we explicitly denote the backbone used for each method to ensure transparency. Crucially, our main comparisons: REPA and REPA-E, employ the exact same SiT backbone as our Send-VAE. All methods are trained on the ImageNet-1k dataset following the standard preprocessing protocol.
> Consequently, the substantial performance improvement of Send-VAE over the previous SOTA (REPA-E) is achieved under an identical training setup (same SiT backbone and data). This confirms that the gains stem from our superior semantic-disentangled VAE design rather than differences in the diffusion framework. We will revise the caption of Table 1 to clarify these settings and prevent ambiguity.
>
> ## Q2 Causal intervention
> We thank the reviewer for the constructive comment.  We emphasize the robust empirical consistency of our findings: Tables 4, 5, and 6 show a strong, systematic correlation between semantic disentanglement and generation performance across diverse VAE architectures, VFMs, and ablations. This consistent trend supports our hypothesis. We are committed to exploring rigorous causal interventions in future work.
>
> ## Q3 Mapper capacity
> We apologize for any confusion regarding the operational phase of the mapper. We would like to clarify that the mapper network is a training-time auxiliary module and is completely discarded during the training of latent diffusion models. Thus, it is impossible for the mapper to compensate for VAE weaknesses at inference time. The performance improvements observed in our experiments are entirely baked into the VAE's weights, confirming that our method indeed produces a VAE with a superior, semantically disentangled latent space.

---

### Author Response · Authors · 2025-12-02

We sincerely appreciate your valuable time and effort spent reviewing our manuscript. Here we have listed the responses to the main questions raised by the reviewers.

## Distinction from VA-VAE

While both methods utilize VFM guidance, Send-VAE differs fundamentally from VA-VAE in motivation, insight, and methodology.
1. Motivation. VA-VAE assumes that the optimization dilemma stems from non-uniform latent distributions and aims to fix this via alignment. We critically re-examine and invalidate this assumption. As shown in Fig. 2 of our paper, while VA-VAE improves uniformity, this metric does not consistently correlate with generation quality. We demonstrate that uniformity is merely a byproduct, necessitating a paradigm shift from geometric regularization to semantic enhancement.
2. Insight. Unlike VA-VAE, we identify Semantic Disentanglement as the true intrinsic predictor of generation-friendliness. Through extensive linear-probing experiments, we reveal that the performance gain stems from distilling semantic structures, not from reshaping the distribution.
3. Methodology. Due to the significant semantic gap between VFMs and VAEs, the simple linear projector used in VA-VAE is insufficient. Send-VAE employs a sophisticated non-linear mapper (and noise injection strategies) designed specifically to bridge this gap. This design enables more effective semantic injection, directly strengthening the VAE's semantic disentanglement ability—a capability that linear alignment fails to fully exploit.

In summary, we contribute by (1) disproving the uniformity hypothesis, (2) establishing semantic disentanglement as a new standard, and (3) proposing specialized architectures (Non-linear Mapper & Noise) that achieve SOTA results by effectively bridging the representation gap.

## Results under 512x512 resolution

We finetune SD-VAE using the proposed $\mathcal{L}_{\text{align}}$ at 512x512 resolution on ImageNet-1k. Then, a Sit-B is trained following the setup in ablation studies. As shown below, at the 512x512 resolution, the proposed method still significantly improves downstream generative performance, demonstrating the effectiveness of our approach. We will subsequently provide more experiments at 512x512 resolution to demonstrate the effectiveness of our method.

| VAE Initialization | gFID$\downarrow$ | sFID$\downarrow$ | IS$\uparrow$ | Prec.$\uparrow$ | Rec.$\uparrow$ |
|------|------|------|------|------|------|
| SD-VAE  | 23.59 | 6.74 | 65.67 | 0.71 | 0.59 |
| +$\mathcal{L}_{\text{align}}$ | 13.32 | 4.75 | 93.15 | 0.78 | 0.60 |

---

### Meta-Review · Area_Chair_Syuy · 2025-12-31

**Summary:**

The critical concerns raised by the reviewers center on two core issues with the proposed Send-VAE method: first, questions regarding its novelty, specifically, that aligning VAE latent space with pre-trained Vision Foundation Model (VFM) representations has already been explored in prior works (e.g., VA-VAE), and that the proposed non-linear mapper is perceived as an incremental architectural adjustment rather than a principled innovation. Second, despite the authors’ detailed rebuttal addressing comparisons with existing methods, there remains a lack of in-depth principled explanations (e.g., theoretical analysis or principled experiments) to precisely interpret the inherent advantages of Send-VAE. Overall, this paper is a borderline case, and the weakness may outweigh the contributions of this paper a bit. Thus, I would recommend rejection.

**Reviewer Concerns:**

The authors partially addressed some reviewer concerns via their detailed rebuttal, including clarifying differences with VA-VAE in motivation, insight and methodology, supplementing experiments at 512×512 resolution, providing ablation results without REPA loss, and disclosing mapper details.

The key remaining issues include justifying the novelty of the paper (the proposed method seems to be distinct from existing ones from the methodology rather than concept), and lack of principled explanations for the method's advantage (currently, the idea seems to be developed from intuition, there lacks rigorous justification and post-hoc explanation, linear probing experiments only show correlation between semantic disentanglement and generation performance, without theoretical support or causal validation to confirm why the non-linear mapper and semantic disentanglement are inherently superior).

**Reviewer Scores:**

Reviewers are not actively participating in the discussion. Based on the authors' rebuttal and the reviewers' previous comments, there is no strong evidence that the reviewers will update their score.

---

### Decision · Program_Chairs · 2026-01-26

Reject